# An Efficient Approach to Prepare Water-Redispersible Starch Nanocrystals from Waxy Potato Starch

**DOI:** 10.3390/polym13030431

**Published:** 2021-01-29

**Authors:** Haijun Wang, Cancan Liu, Runyan Shen, Jie Gao, Jianbin Li

**Affiliations:** 1School of Light Industry and Food Engineering, Guangxi University, Nanning 530004, China; 18856329582@163.com (H.W.); liucancan2919@163.com (C.L.); 18838919087@163.com (R.S.); 2Guangxi Key Laboratory of Biorefinery, Guangxi Academy of Sciences, Nanning 530003, China

**Keywords:** starch nanocrystals, waxy potato starch, water redispersibility, dispersion stability

## Abstract

Starch nanocrystals (SNCs) are a biodegradable polymer which has been widely studied and used in many fields. In this study, we have developed an efficient procedure for the preparation of SNCs. First, sodium hexametaphosphate (SHMP) and vinyl acetate (VAC) were used to modify waxy potato starch (WPS). Then, the modified starches were hydrolyzed with sulfuric acid to prepare SNCs. Results showed that SNCs prepared with modified starch had higher zeta potentials and better dispersion properties than the original starch. After modification, WPS still maintained its semi-crystalline structure, but the surface became rougher. SHMP-modified WPS showed a decrease in viscosity peak and an increase in gelatinization temperature. VAC-modified WPS showed increased swelling power. Additionally, SNCs prepared with VAC-modified WPS had better water redispersibility and dispersion stability than those from SHMP-modified starch—which will have broader application prospects in the field of safe and biodegradable food packaging.

## 1. Introduction

Starch is an abundant biopolymer which comes mainly from grains and tubers and is totally biodegradable [1,2,3]. The preparation and utilization of starch nanocrystals (SNCs) originating from starch granules has been extensively studied for many years [4,5,6,7]. The preparation and physical properties of SNCs have been the subject of great interest and enthusiastic study in recent years [8,9,10]. SNCs are the crystalline fragments that result after enzymatic hydrolysis or acid hydrolysis of the amorphous areas of starch [11]. Compared to general starches, SNCs have smaller size (nanoscale), higher crystallinity, a regular thin sheet structure, low permeability, and strong digestive resistance. Starch nanocrystals have been widely studied and used as emulsion stabilizers [12,13,14], membrane-reinforcing fillers [15,16], and even rubber-reinforcing fillers [17], due to their excellent characteristics. However, SNCs tend to aggregate and settle in water [18]. In order to overcome the poor dispersion properties of SNCs in solvents, chemical modification methods have been used to improve the dispersibility of SNCs in water and organic solvents [11,18,19,20]. For these methods, the starch is first hydrolyzed with sulfuric acid and then chemically modified. After being hydrolyzed by sulfuric acid, the starch needs to be washed to neutral by distilled water. After chemical modification, the washing step needs to be performed again. Due to the extremely small particle size of SNCs, the washing step becomes more cumbersome and there will be more losses at this stage. In order to improve the preparation efficiency of SNCs and reduce loss during the preparation process, it is suggested to conduct chemical modification before sulfuric acid hydrolysis to prepare SNCs with better water redispersibility.

Waxy potato starch has been widely studied because of its many excellent characteristics [21]. Waxy potato starch is derived from potato mutants. Compared with general corn starch and tapioca starch, waxy potato starch has higher amylopectin content. Studies have shown that high amylose content will hinder acid hydrolysis to a certain extent, and that starches with lower amylose content have smaller SNC sizes and higher crystallinity. Waxy potato starch (WPS) contains 99% amylopectin and is very suitable for producing SNCs [22,23,24]. There are many reports on starch modification, but few reports on the modification of waxy potato starch with sodium hexametaphosphate (SHMP) and vinyl acetate (VAC) and the preparation of SNCs from modified starch. Both SHMP and VAC can be used in the food industry. SHMP is used in meat products as an additive to improve taste and inhibit oxidation [25]. The VAC polymer can be used in chewing gum [26]. Therefore, it is relatively safe to use SHMP and VAC in the modification of starch.

The purpose of this research was to obtain SNCs with good dispersing properties through the method of, first, modification and then sulfuric acid hydrolysis. Herein, the characteristic changes of the modified starches were evaluated by attenuated total reflection–Fourier transform infrared spectroscopy (ATR–FTIR), scanning electron microscopy (SEM), thermal stability, adhesion property, gelatinization, and swelling force tests. The structural characteristics of the modified starches were determined by X-ray diffraction and polarizing microscopy. The modified starch nanocrystals were characterized by Marvin particle size (zeta potential) measurements and wettability tests.

## 2. Materials and Methods

### 2.1. Materials

Waxy potato starch (WPS) was kindly provided by the Avibe Company (Avibe, The Netherlands). SHMP was purchased from Shanghai Hongshun Biotechnology Co., Ltd. (Shanghai, China). Sodium chloride, anhydrous sodium carbonate, and VAC were purchased from Shanghai Alighting Biochemical Technology Co., Ltd. (Shanghai, China). Sulfuric acid (≥95%) was purchased from Beijing Chemical (Beijing, China).

### 2.2. Modification of Waxy Potato Starch

For modification by SHMP, 40 g of WPS was added to 100 mL deionized water and stirred magnetically at 45 °C for 10 min. A total of 1 g Na_2_CO_3_, 2.5 g NaCl, and different amounts of SHMP (4 wt%, 6 wt% and 8 wt% of WPS) were added to the starch suspension [18]. The pH was adjusted to 10 with 1 mol/L NaOH, and the mixture was stirred continuously for 4 h at 45 °C. After the reaction was complete, the starch suspension was adjusted to pH 6.8 with HCl (1 mol/L), washed with deionized water, filtered three times, washed with anhydrous ethanol three times, dried at 40 °C, sifted, and stored [18].

For VAC modification, a 40% suspension of WPS was prepared, and 1 g Na_2_CO_3_ was added as a buffer and stabilizer [27]. Using a dilute alkali solution (1 mol/L of NaOH), the starch suspension was adjusted to pH 10 and constantly stirred. Then, different amounts of VAC (4 wt%, 6 wt% and 8 wt% of WPS) were added drop by drop [28,29], and the temperature was raised to 40 °C for 2 h. After the reaction was complete, the starch slurry was adjusted to pH 6~7, washed four times with 70% ethanol, and washed three times with anhydrous ethanol. The sample was dried at 40 °C, sifted, and stored.

### 2.3. Preparation of Starch Nanocrystals (SNCs)

Waxy potato SNCs were obtained by sulfuric acid hydrolysis of waxy maize starch according to the method of Angellier et al. [30]. Briefly, 29.4 g of modified potato starch granules was dispersed in 200 mL of H_2_SO_4_ solution (3.16 mol/L) at 40 °C and stirred at 200 rpm for 6 days. Then, the nanocrystals were separated from the acid by centrifugation at 10,000 rpm at 25 °C for 10 min. The deposits were washed in distilled water until the pH of the supernatant was constant. Finally, SNC powder was obtained by lyophilizing the neutralized suspension.

### 2.4. Sample Characterization

The absorption spectra of the samples were recorded using attenuated total reflection–Fourier transform infrared (FTIR) spectrometry (TENSOR II, Bruker, Karlsruhe, Germany) at a resolution of 4000–400 cm^−1^ [31]. Scanning electron microscopy (SEM) (s-3400n, Hitachi, Tokyo, Japan) was used to study the morphology of the sample at 5 kV acceleration voltage. A polarizing microscope (BX41, Olympus, Tokyo, Japan) was used to observe the sample under cross-polarized light (200× magnification). The crystal structure of the sample was measured with an X-ray diffractometer (Smartlab 3KW, Rigaku, Tokyo, Japan) from 4° to 40° at a rate of 5°/min, with a step length of 0.02°. Thermogravimetric (TG) and derivative thermogravimetric (DTG) analyses were performed using a STA449-F5TAQ600 thermal analyzer (NETZSCH, Waldkraiburg, Germany) under a nitrogen flow of 20 mL/min. The sample (approximately 10 mg) was heated from 30 to 600 °C at a rate of 10 K/min. The adhesive properties of the modified starch samples were assessed using a Viscograph-E (Brabender, Duisburg, Germany) [32]. The particle size (zeta potential) of the sample was determined using a nano-zeta potential analyzer (Nano-ZS90, Malvern Panalytical, Malvern, UK). The measurements were performed using the samples prepared by dispersing starch nanocrystals in deionized water at 25 °C at a ratio of 0.01% *w/v*.

### 2.5. Swelling Power

The swelling power was determined following the method described by Qian et al. [33]. The 2% (*w/w*) starch sample suspension was kept at 40, 50, 70, or 90 °C for 30 min, then cooled to room temperature and centrifuged at 10,000 rpm for 10 min. The supernatant was discarded, and the precipitate was weighed. The swelling power (g/g based on dry weight) was calculated as follows:(1)$$\mathrm{swelling}\;\mathrm{power}\left(\frac{g}{g}\right)=\frac{\mathrm{sediment}\;\mathrm{weight}}{\mathrm{mass}\;\mathrm{of}\;\mathrm{dry}\;\mathrm{starch}\;\times \;\left(1-\mathrm{solubility}\right)}$$

### 2.6. Wettability Tests

The SNCs were dispersed in deionized water at a concentration of 3 mg/mL following the method described by Zhou et al. [34]. The dispersion of SNCs in deionized water was recorded at different time points to evaluate the dispersibility of the samples.

### 2.7. Statistical Analysis

Experiments were performed in triplicate, and the data were represented as the average value ± standard deviation. The difference between factors and levels was evaluated by analysis of variance (ANOVA). Duncan tests were conducted to examine differences among experimental mean values, and *p* < 0.05 was considered significant.

## 3. Results and Discussion

### 3.1. FTIR Analysis

The FTIR spectrum of waxy potato starch modified by hexametaphosphate is shown in Figure 1A. There was a characteristic absorption peak near 2926 cm^−1^, which was attributed to stretching and antisymmetric stretching of the C–H bond [22]. The absorption peak near 1642 cm^−1^ was attributed to bending vibrations of H–O–H [35]. The absorption peaks near 995 cm^−1^ and 1150 cm^−1^ were caused by stretching vibrations of C–O [36]. The absorption peak at 1078 cm^−1^ was caused by stretching vibrations of O–C in the C–O–C group of the glucose ring [37,38]. According to the analysis of Ren et al., the absorption peak of P=O vibrations is relatively weak at 1400–1150 cm^−1^ [18]. There was a change in the absorption peak of the modified starch near 1298 cm^−1^, indicating that the starch was indeed modified. The absorption peak of the P–O–C vibration, located at 1050–995 cm^−1^, overlapped with the absorption peak of C_6_–OH vibrations in the starch and glucose units, which was difficult to directly observe by ATR–FTIR. It has also been suggested that the phosphoric acid cross-linking reaction peak at about 3300 cm^−1^ was due to migration of the stretching peak of the hydroxyl group [39]. The IR bands at around 1047 and 1022 cm^−1^ were related to the crystalline and amorphous areas of starch, respectively, and the ratio of absorbances at 1047 and 1022 cm^−1^ was usually used to characterize the short-range molecular order of starch [40,41]. The absorbance ratios of 1047/1022 cm^−1^ of native and modified WPS with 4% SHMP, 6% SHMP, and 8% SHMP were 0.6367, 0.6202, 0.6349, and 0.6215, respectively. The results showed that the short-range molecular sequence of the modified starch was destroyed by the cross-linking reaction. This also illustrated the occurrence of modification.

The ATR–FTIR spectrum of the waxy potato starch modified by VAC is shown in Figure 1B. The wide absorption peak at 3303 cm^−1^ represented stretching vibrations of O–H [42]. The absorption peak at 1643 cm^−1^ was attributed to bending vibrations of H–O–H in combined water [43]. The absorption peak near 998 cm^−1^ represented stretching vibrations of C–O. At 1724 cm^−1^, a new absorption peak was caused by stretching vibrations of C=O [28], owing to the introduction of new groups via the successful reaction between VAC and starch. Due to the introduction of these new groups, the absorption peak also changed at 2892 cm^−1^ and 2832 cm^−1^, which was attributed to interactions with C–H vibrations [22]. The absorbance ratios of 1047/1022 cm^−1^ of native and modified WPS with 4% VAC, 6% VAC, and 8% VAC were 0.6460, 0.6244, 0.6274, and 0.6288, respectively. This also illustrated the occurrence of modification.

### 3.2. Scanning Electron Microscopy (SEM)

The SEM micrographs of native and modified WPS with SHMP are presented in Figure 2. As shown in Figure 2a’, micropores were found on the surface of the starch granules, reaching from the surface to the umbilical point, the formation of which was related to the formation of starch granules. The starch grains grew outward from the umbilical point in a regular radial lamellar structure from the spherulites. Due to the loose aggregation of defective microcrystals, this area was easily shed, forming pores on the surface of the starch particles [44]. After modification of the waxy potato starch with different concentrations of SHMP, the overall shape of the starch granules changed little, but the surface roughness increased, which may have been caused by cross-linking of the modifier with the hydroxyl group on the surface of the starch. During the cross-linking reaction between starch and SHMP, SHMP reacted with the hydroxyl groups on starch molecules, forming intra- and inter-ester linkages which cross-linked starch molecules [45]. Moreover, the starch granules adhered firmly to each other due to the cross-linking reaction between starch and SHMP [41].

The morphologies of the acetylated waxy potato starch and the original starch are shown in Figure 3. Potato starch had an oval appearance and a smooth surface. The acetyl in modified waxy potato starches may have increased the hydrogen bonding, resulting in the fusion of starch granules [29]. Observations at 1000× and 5000× magnification showed that the surface of the potato starch became rough after modification, and various shapes of modified starch were observed. Vinyl acetate was mainly grafted and polymerized at the C2 position [46], which made the surface uneven. Holes were also observed on the surface of the starch, which may be caused by the reaction of vinyl acetate entering the starch [27]. 

### 3.3. Polarizing Microscopy Analysis

Starch has a semi-crystalline structure, and the amorphous and semi-crystalline regions are arranged alternately. The amorphous and semi-crystalline regions differ in density and refractive index, and cross-polarization occurs when polarized light passes through the starch granules [47]. The entire starch granule exhibited birefringence under a polarized light microscope, and a polarized light cross could be seen clearly [48]. The intensity of birefringence of the starch granules depends on their radius and the size of their helix structure [44]. Figure 4a–d shows polarizing microscopy images of starch samples before and after modification by SHMP. After modification of the starch with different amounts of modifier, the polarization cross was still clearly visible. The modification method did not change the semi-crystalline structure of the starch but had some effect on the surface of the starch, indicating that the modification occurred on the surface of the starch. Figure 4e–g shows polarizing microscopy diagrams of starch samples before and after modification by VAC. As the concentration of the modifier increased, part of the granules appeared to develop cavities. However, the structure of cross-polarized crosses outside still existed. As indicated by the mark in Figure 4g, only part of the cross-center structure of the particles was weakened. This may be due to the fact that VAC enters internal regions of starch granules that have a less compact structure, resulting in only slight structural changes.

### 3.4. X-ray Diffraction (XRD)

The X-ray diffraction patterns of WPS modified with SHMP (4 wt%, 6 wt%, and 8 wt% of WPS) are shown in Figure 5A. WPS displayed characteristic peaks at 2 = 5.6°, 17.2°, 22.2°, and 24.0° with typical B-type crystal structures [43]. After modification, the intensity of each peak was decreased, especially that at 2 = 5.6°, and the peaks around 15° also gradually narrowed. The crystallinity of the modified starch was decreased compared with that of the original starch, but the B-type crystal structure of the starch was not changed. The cross-linking reaction between WPS and SHMP occurred mainly on the surface of the starch granules.

X-ray diffraction patterns of WPS modified with VAC (4 wt%, 6 wt%, and 8 wt% of WPS) are shown in Figure 5B. The original WPS exhibited characteristic peaks at 2 = 5.6°, 17.2°, 22.2°, and 24.0° with typical B-type crystal structures [43]. After modification, the intensity of the peaks at 2 = 17.2°, 22.2°, and 24.0° was significantly reduced, and the peaks around 15° gradually narrowed. The crystallinity of the modified starch was reduced compared with that of the original starch, but the B-type crystal structure of the starch did not change. As the size of the VAC-modified starch increased, the decrease in crystallinity may have been due to an increase in the proportion of the amorphous region caused by grafting polymerization on the surface of the starch particles.

### 3.5. Thermogravimetric Analysis 

TG and DTG curves of WPS and its modified starch are shown in Figure 6. As shown in Figure 6A,C, the graphs revealed that there were two stages of thermal degradation common to all heating rates. In the first stage, there was weight loss of 10.3% to 13.0% from 30 to 250 °C, mainly due to the evaporation of the existing intramolecular and intermolecular moisture [49,50,51]. The second stage of degradation consisted of sharp degradation, which took place from 250 °C to 350 °C and extends further up to about 620 °C. The weight loss that occurred in this stage was caused mainly by breaking of the starch carbon chain, decomposition of glucose molecules, and the rapid carbonization of the starch [52]. As shown in Figure 6B, the decomposition rate of SHMP-modified starch at the maximum decomposition temperature was reduced, which was attributed to the fact that the cross-linking reaction increased the compactness of the starch structure, and the compact structure led to restriction of movement of the molecular chains and increased resistance to decomposition [41,53]. As shown in Figure 6D, the maximum degradation temperature of VAC-modified starch increased, originating from the interaction between the hydroxyl groups on the SNCs surface and acetate groups of VAC [37].

### 3.6. Swelling Power

Figure 7 shows the swelling power at different temperatures of starches modified by the two different methods. When the gelatinization temperature of the WPS was lower than that of the gelatinization temperature, the swelling force of the VAC-modified starch increased with temperature. The larger the amount of modifier, the more obvious the increase [29]. The swelling power of the SHMP-modified starch did not change significantly. In the case of higher gelatinization temperatures, the change in modified starch was smaller and the swelling force was lower than that of unmodified starch, but the swelling force of VAC-modified starch was higher than that of SHMP-modified starch, which was mainly attributable to the hydrophilic water retention capacity of VAC. According to the research of Kaur et al. [54], cross-linking modifications to SHMP greatly reduce its swelling force, water binding ability, and paste transparency. The reduced swelling of modified VAC at higher temperatures than that of the original starch was attributed to reduced gelatinization temperature [27].

### 3.7. Pasting Properties

The characteristic parameters of all samples are shown in Table 1. Both modification methods changed the viscosity of the starch. The values of peak viscosity (PV) for crosslinked starches were significantly lower than those of native starches. PV also decreased with an increase in the level of cross-linkage. The decrease in PV of cross-linked starches may be due to formation of cross-linked covalent bonding between amylose and amylopectin chains, which result in the strengthening of swollen starch granules, thereby resulting in lower breakage of paste under thermal and mechanical shear [55]. PV of VAC-modified (4 wt% and 6 wt% of WPS) starch increased and gelatinization temperature decreased, owing to its strong water absorption properties [27]. However, VAC-modified (8 wt% of WPS) WPS gelatinized more easily owing to its low gelatinization temperature.

### 3.8. Zeta Potentials and Size Distributions of SNCs

Table 2 shows the zeta potential and Z-average size of SNCs prepared from original starch and two types of modified starch. SNCs have a wide particle size distribution, mainly due to aggregation [34]. The particle size of SNCs obtained by the acidolysis of SHMP- and VAC-modified starch was higher than that of SNCs obtained by the acidolysis of the original starch. A zeta potential of 0.01% (*w/v*) SNCs was measured at pH 7.0 to evaluate the water dispersion stability of the prepared SNCs. The higher the absolute value of the zeta potential, the greater the stability of the dispersion system [56]. According to the results, the absolute value of the zeta potential of SNCs prepared from modified starch was higher than that of the original waxy potato starch, indicating that they had higher dispersion stability in water. Because the surface of SNCs prepared from denatured starch contained more negatively charged groups, a repulsive force existed that could prevent particle aggregation.

### 3.9. Wettability Experiments

Figure 8 shows SNCs from the original and modified starch uniformly dispersed in deionized water at a concentration of 3 g/L. Figure 8A–D shows the changes in dispersion observed at 0 h, 2 h, 12 h, and 24 h, respectively. The SNCs prepared from the SHMP-modified (4 wt% and 6 wt% of WPS) starch showed relatively high sedimentation after 2 h, which almost completely subsided after 12 h; the remaining SNCs were still dispersed after 24 h. Because the SNCs prepared by the waxy potato starch modified by SHMP have a larger particle size, the SNCs with a low degree of substitution settled at an earlier time. The presence of carboxylates and sulfates on the surface of SNCs obtained by hydrolysis of WPS with sulfuric acid gives them a certain degree of dispersibility [57]. SNCs prepared with SHMP- (8 wt% of WPS) and VAC-modified (8 wt% of WPS) WPS maintained good dispersion after 12 h, and SNCs prepared with VAC-modified starch maintained good dispersion after 24 h. The dispersion of SNCs corresponded to their zeta potentials. Therefore, we concluded that the modification of waxy potato starch with SHMP or VAC improved the dispersibility of SNCs, and that the effects of VAC modification were superior.

## 4. Conclusions

The WPS was chemically modified and then hydrolyzed with sulfuric acid to obtain SNCs, which had higher dispersibility than those obtained by direct sulfuric acid hydrolysis of native starch. This method simplified the preparation steps of SNCs. It can be concluded from the X-ray diffraction and polarized light microscopy that both SHMP and VAC modifications did not change the crystal configuration of waxy potato starch, but the relative crystallinity was decreased. Starch modified with SHMP had lower peak viscosity and was difficult to gelatinize. VAC-modified waxy potato starch had increased swelling. SNCs prepared with modified starch had higher zeta potentials and better dispersion properties than the original starch. The dispersion of SNCs modified with VAC was better than that of SNCs modified with SHMP. Additionally, SNCs prepared with VAC-modified (8 wt% of WPS) WPS had the best water dispersion properties and dispersion stability. This method of preparing SNCs by starch modification and hydrolysis provided new possibilities for improving the dispersibility of SNCs and may be used in the field of safe and biodegradable food packaging.

## Figures and Tables

**Figure 1 polymers-13-00431-f001:**
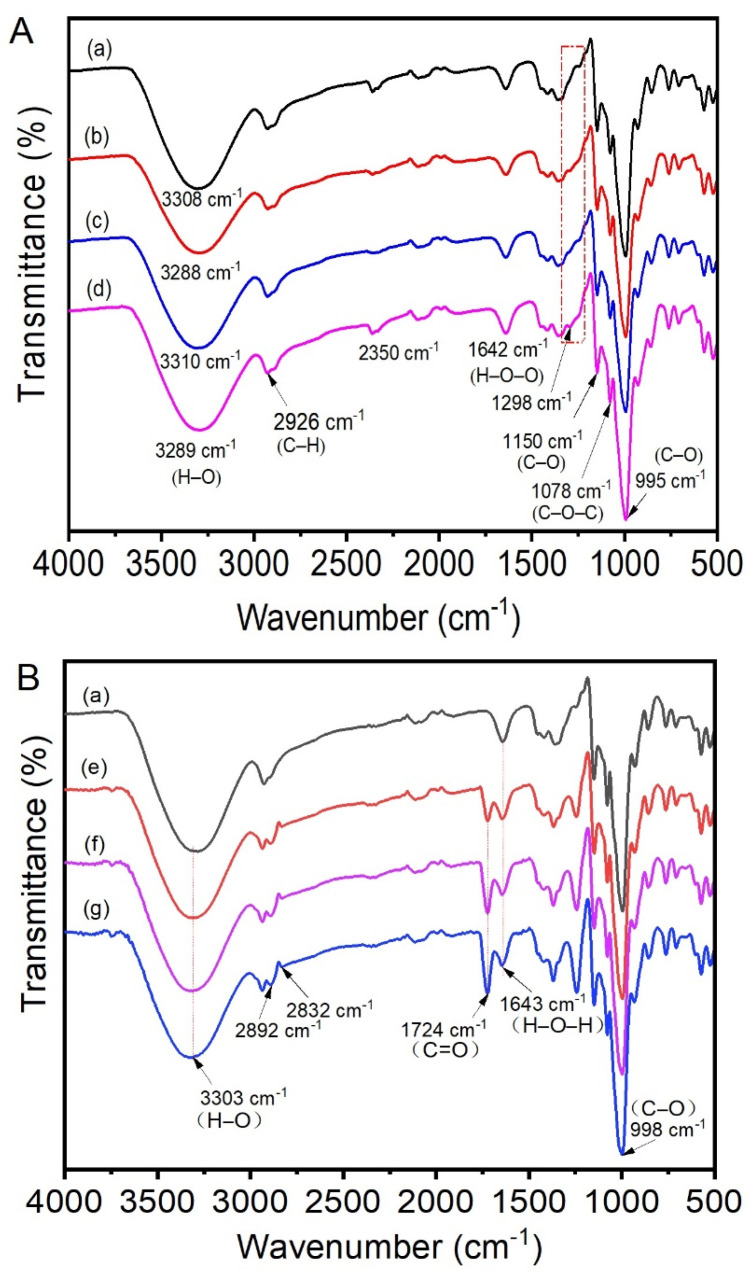
(**A**) Attenuated total reflection– Fourier transform infrared spectroscopy (ATR–FTIR) spectra of native (a) and modified waxy potato starch (WPS) with 4% sodium hexametaphosphate (SHMP) (b), 6% SHMP (c), and 8% SHMP (d), (**B**) ATR–FTIR spectra of native (a) and modified WPS with 4% vinyl acetate (VAC) (e), 6% VAC (f), and 8% VAC (g).

**Figure 2 polymers-13-00431-f002:**
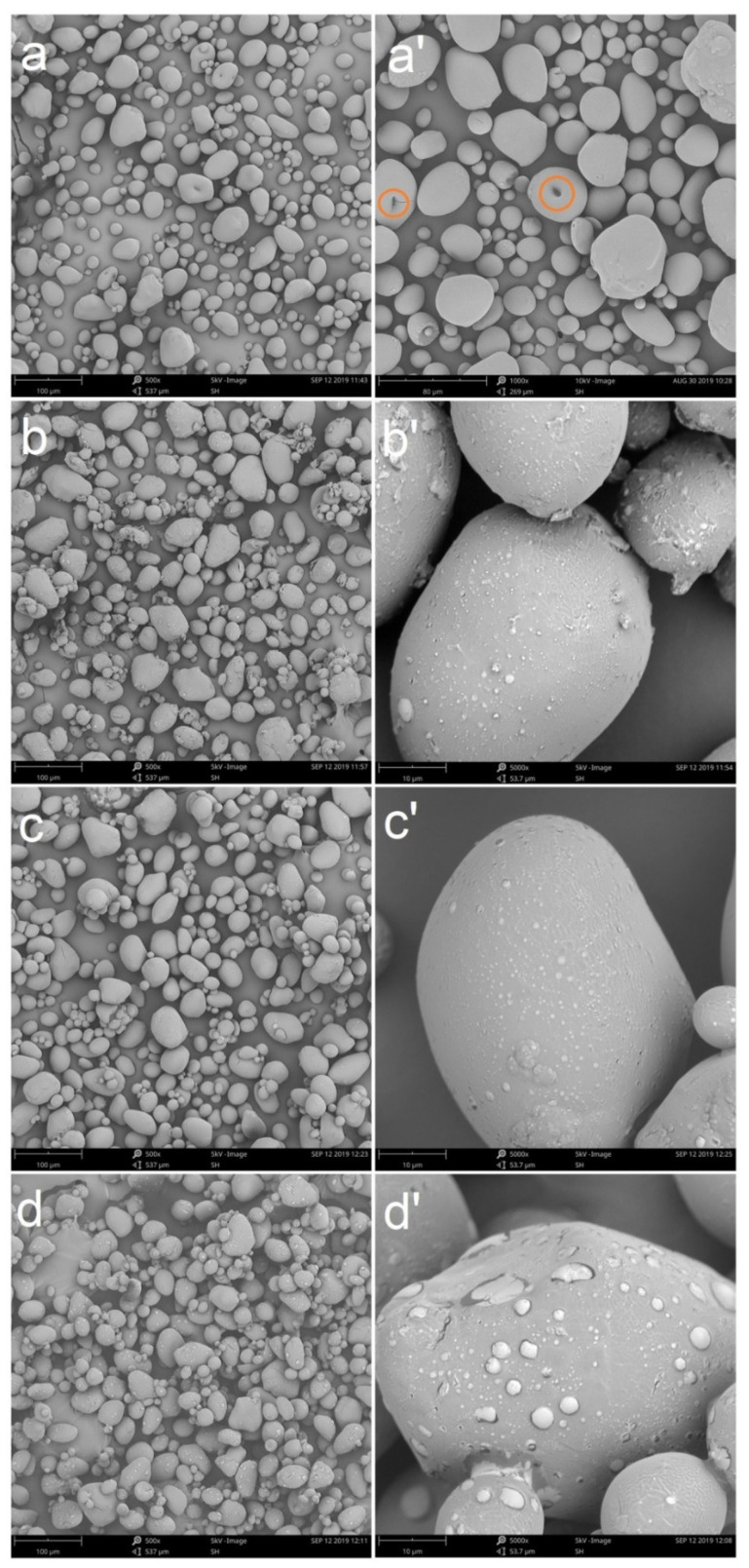
Scanning electron microscopy (SEM) micrographs of native (**a**,**a’**) and modified WPS with 4% SHMP (**b**,**b’**), 6% SHMP (**c**,**c’**) and 8% SHMP (**d**,**d’**) (micrographs of each starch sample were taken at 500× and 5000× magnification).

**Figure 3 polymers-13-00431-f003:**
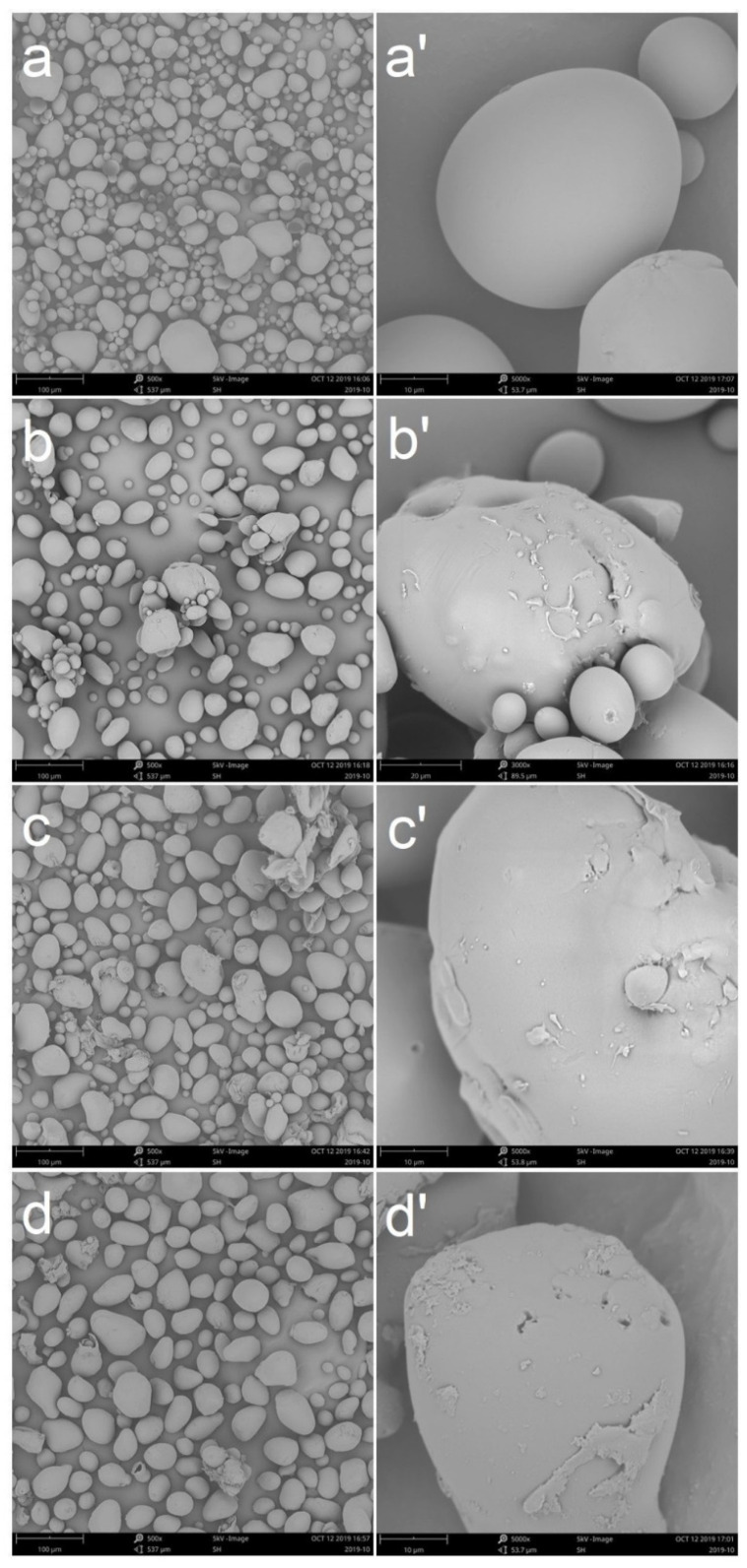
SEM micrographs of native (**a**,**a’**) and modified WPS with 4% VAC (**b**,**b’**), 6% VAC (**c**,**c’**), and 8% VAC (**d**,**d’**) (micrographs of each starch sample were taken at 500× and 5000× magnification).

**Figure 4 polymers-13-00431-f004:**
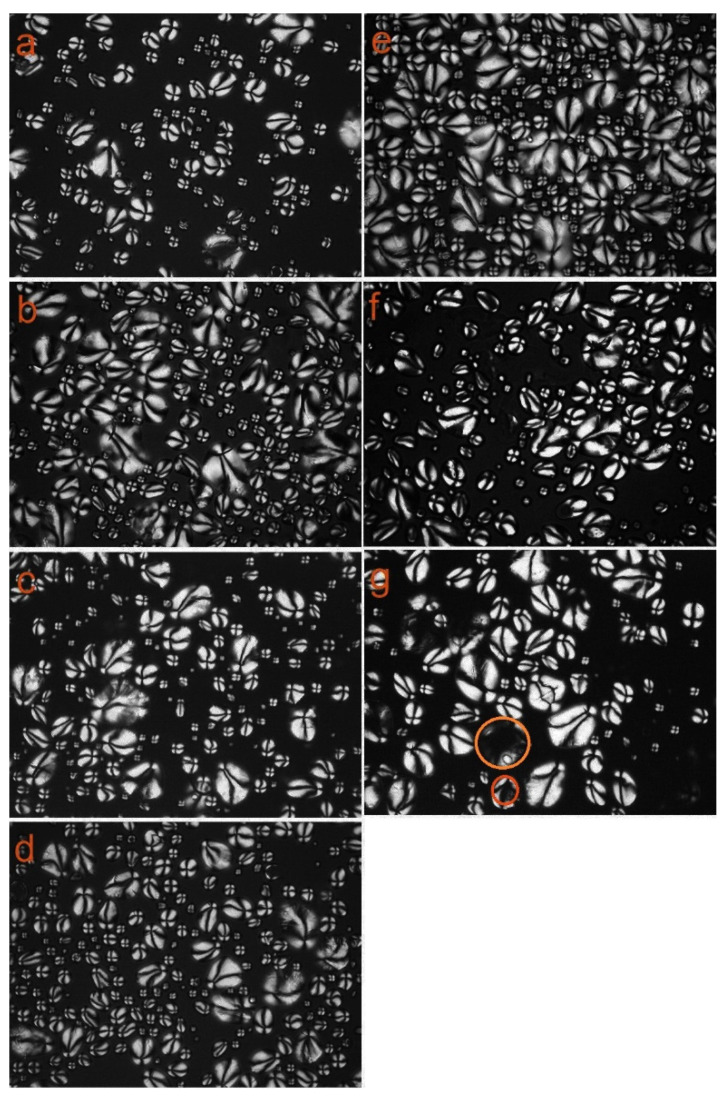
Polarizing micrograph (500×) of native (**a**) and modified WPS with 4% SHMP (**b**), 6% SHMP (**c**), 8% SHMP (**d**), 4% VAC (**e**), 6% VAC (**f**), and 8% VAC (**g**).

**Figure 5 polymers-13-00431-f005:**
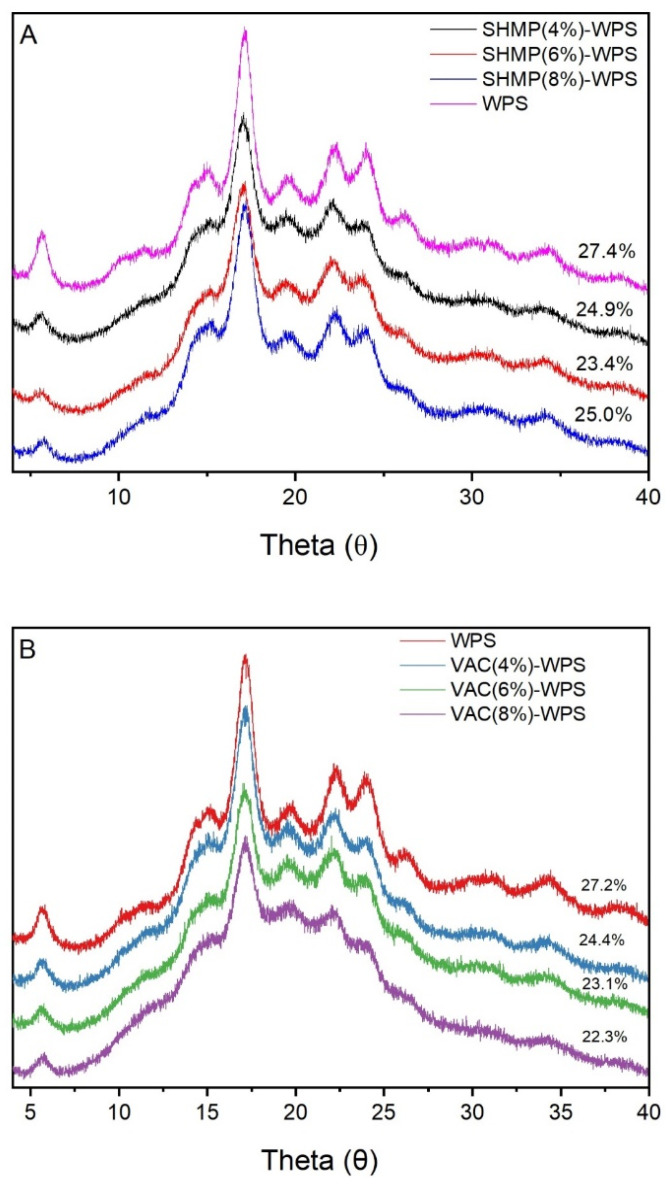
(**A**) The X-ray diffraction patterns of native and modified WPS with 4% SHMP, 6% SHMP, and 8% SHMP, (**B**) The X-ray diffraction patterns of native and modified WPS with 4% VAC, 6% VAC, and 8% VAC.

**Figure 6 polymers-13-00431-f006:**
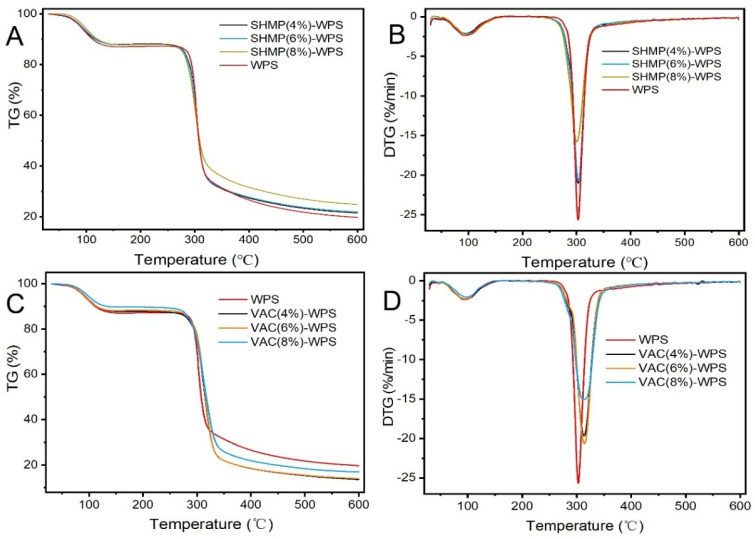
(**A**)Thermogravimetric (TG) and (**B**) derivative thermogravimetric (DTG) analyses of native and modified WPS with 4% SHMP, 6% SHMP, 8% SHMP; (**C**) TG and (**D**) DTG analyses of native and modified WPS with 4% VAC, 6% VAC, and 8% VAC.

**Figure 7 polymers-13-00431-f007:**
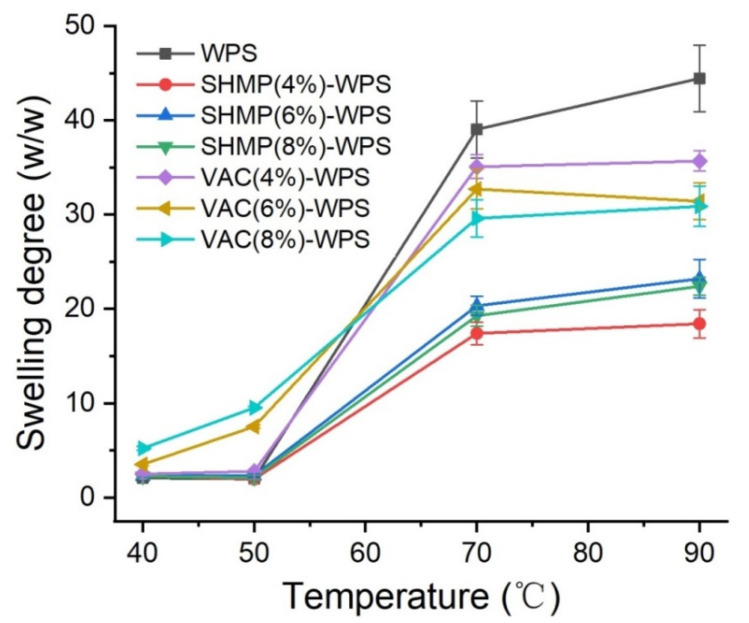
Swelling power of native and modified WPS with 4% SHMP, 6% SHMP, 8% SHMP, 4% VAC, 6% VAC, and 8% VAC.

**Figure 8 polymers-13-00431-f008:**
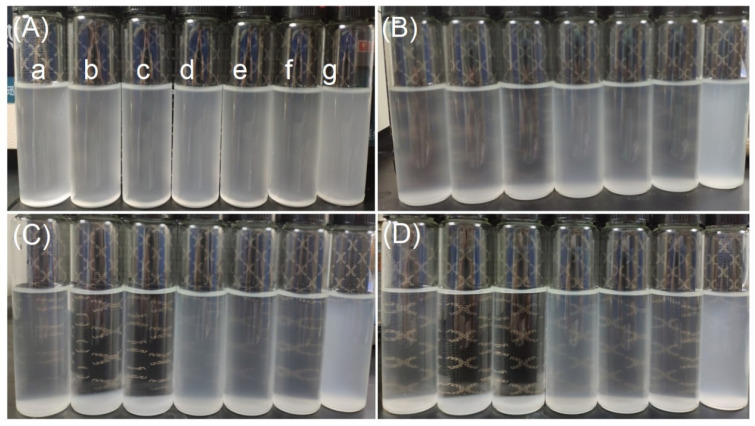
Dispersion of starch nanocrystals (SNC) in water: 0 h later (**A**), 2 h later (**B**), 12 h later (**C**), 24 h later (**D**); SNCs prepared by native (a) and modified WPS with 4% SHMP (b), 6% SHMP (c), 8% SHMP (d), 4% VAC (e), 6% VAC (f), and 8% VAC (g).

**Table 1 polymers-13-00431-t001:** Pasting properties of native, VAC, and SHMP modified WPS.

Sample	Pasting Temperature (°C)	Peak Viscosity (BU)
WPS	65.6 ± 0.4 ^c^	1312.0 ± 101.3 ^b^
SHMP(4%)-WPS	67.3 ± 0.3 ^b^	851.0 ± 50.4 ^d^
SHMP(6%)-WPS	67.3 ± 0.4 ^b^	686.0 ± 23.1 ^e^
SHMP(8%)-WPS	68.6 ± 0.2 ^a^	535.0 ± 30.2 ^f^
VAC(4%)-WPS	57.9 ± 1.1 ^b^	1539.0 ± 70.4 ^a^
VAC(6%)-WPS	54.6 ± 0.5 ^d^	1611.0 ± 80.6 ^a^
VAC(8%)-WPS	53.7 ± 0.6 ^e^	1284.0 ± 45.7 ^c^

^a–f^ Mean values with different lowercase letters in the same row are significantly different (*p* < 0.05).

**Table 2 polymers-13-00431-t002:** Zeta potential of native, VAC-, and SHMP-modified WPS.

Sample	Z-Average (nm)	Zeta Potential (mV)
WPS	272.0 ± 9.5 ^a^	−11.1 ± 0.2 ^e^
SHMP(4%)-WPS	276.8 ± 11.2 ^a^	−15.7 ± 0.3 ^d^
SHMP(6%)-WPS	580.6 ± 40.1 ^c^	−17.4 ± 0.4 ^c^
SHMP(8%)-WPS	392.1 ± 17.1 ^b^	−18.4 ± 0.3 ^b^
VAC(4%)-WPS	385.4 ± 15.6 ^b^	−18.1 ± 0.1 ^b^
VAC(6%)-WPS	548.1 ± 35.3 ^c^	−17.3 ± 1.0 ^c^
VAC(8%)-WPS	383.8 ± 20.1 ^b^	−20.1 ± 0.2 ^a^

^a–e^ Mean values with different lowercase letters in the same row are significantly different (*p* < 0.05).

## Data Availability

The data presented in this study are available on request from the corresponding author.

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
