# Peer review of "An Efficient Approach to Prepare Water-Redispersible Starch Nanocrystals from Waxy Potato Starch"

_polymers, 2021, doi:10.3390/polym13030431_

Round 1
Reviewer 1 Report
This work is novel and interesting. In addition, it is in the scope of this journal. So, I recommed its publication after performing the following suggestions:
-Line 11: What mean the respectively? Do you mean that two different modifications have been made? It's confusing.
-Line 15: Change peak viscosity for viscosity peak
-Line 17: Change the best for better
-The use of modification as keyword is very generic. You can omit it.
-The most used sources to obtain starch could be defined in the introduction.
-Line 26: Explain better the last investigations of these studies
-Change "WPS containing 99% amylopectin is very suitable to produce SNCs" for "WPS contains 99% amylopectin, being very suitable to produce SNCs"
-What is the solubility of WPS? Does it change with the pH modification? Has it been taken in consideration?
-Why different temperatures and stirred times were selected for SHMP and VAC? Is it due to previous studies or reference works?
-Line 166: Rewrite this sentence: "The morphology of native starch, SHMP-crosslinked starch as observed under SEM 166 are presented in Fig. 2." It is a bit confusing.
-Line 207-208: "but had some ef-207 fect on the surface of starch" As which?
-The results and conclusions obtained are very interesting
Author Response
Thanks so much for the reviewer’s kind suggestion and comments. According to your instructive suggestion, this manuscript has been revised carefully. The followings are our point-by-point responses.
- Line 11: What mean the respectively? Do you mean that two different modifications have been made? It's confusing.
Response: We really appreciate and agree with the reviewer’s comments. The “respectively” was used to mean two different modification treatments for waxy potato starch, which really make it more confusing. So it has been deleted at line 11 in the new manuscript.
- Line 15: Change peak viscosity for viscosity peak.
Response: As suggested by the reviewer, We have changed "peak viscosity" to "viscosity peak" at line 15.
- Line 17: Change the best for better.
Response: We agree with the reviewer’s advice and have altered "best" to "better" at line 17.
- The use of modification as keyword is very generic. You can omit it.
Response: The keyword “modification” was deleted.
- The most used sources to obtain starch could be defined in the introduction.
Response: According to your suggestions, we have added relevant content in the introduction at lines 23-24.
- Line 26: Explain better the last investigations of these studies.
Response: According to the Reviewer's comments, we have added the latest investigations of these studies at line 25-27 and added new references in the manuscript.
- Change "WPS containing 99% amylopectin is very suitable to produce SNCs" for "WPS contains 99% amylopectin, being very suitable to produce SNCs".
Response: We have changed it to “WPS contains 99% amylopectin, being very suitable to produce SNCs” at line 49.
- What is the solubility of WPS? Does it change with the pH modification? Has it been taken in consideration?
Response: The solubility of WPS is about 4% (Jang H S, Lee J, Lee H J, et al. Phytate-mediated phosphorylation of maize, rice, and potato starches at different pH conditions[J]. International Journal of Biological Macromolecules, 2020, 165: 857-864.). As the pH increased, the solubility of WPS decreased (Liu K, Hao Y, Chen Y, et al. Effects of dry heat treatment on the structure and physicochemical properties of waxy potato starch[J]. International journal of biological macromolecules, 2019, 132: 1044-1050.). Considering the very low solubility of WPS, we haven’t taken it into consideration.
- Why different temperatures and stirred times were selected for SHMP and VAC? Is it due to previous studies or reference works?
Response: The different temperatures and stirring times have been selected according to the previous research as listed in the references [18, 27-30].
- Line 166: Rewrite this sentence: "The morphology of native starch, SHMP-crosslinked starch as observed under SEM 166 are presented in Fig. 2." It is a bit confusing.
Response: The statements has been corrected to “The SEM micrographs of native and modified WPS with SHMP were presented in Fig. 2” at line 167.
- Line 207-208: "but had some effect on the surface of starch" As which?
Response: From the results of SEM micrographs, we could see that the surface of modified starch became more roughness. So, we concluded that the modification method had some effect on the surface of starch.
- The results and conclusions obtained are very interesting
Response: Special thanks to you for your good comments.
Reviewer 2 Report
Recommendation: accept
Comments to Authors: Haijun Wang, Cancan Liu, Runyan Shen, Jie Gao, Jianbin Li
Manuscript Number: polymers-1088288-peer-review-v1
Article Type: Article
Article Title: An efficient approach to prepare water-redispersible starch nanocrystals from waxy potato starch
Overview and general recommendation
The introduction provides a sufficient background and contains relevant references to the problem raised.
The methods were presented correctly and chronologically adequate to the conducted research and are adequately described.
The research design is appropriate.
Minor comments:
- please support conclusions by the results;
.
Overall Recommendation
My recommendation is accept
Author Response
Thanks so much for the reviewer’s kind suggestion and comments. According to your instructive suggestion, this manuscript has been revised carefully. The followings are our point-by-point responses.
- Minor comments: Please support conclusions by the results
Response: We have re-written the conclusion in the manuscript according to your suggestion.
- Overall Recommendation: My recommendation is accept.
Response: Special thanks to you for your good comments.
Reviewer 3 Report
In this study, authors obtained SNCs with good dispersing properties 55 through the method of first modification and then sulfuric acid hydrolysis. The manuscript is well-prepared but some comments are linked below.
Materials and methods section:
- Line 125: Why authors used the Duncan test instead of Tukey one?
Results and discussion section:
- Lines 131-132: please add reference after H-O-H.
- Table 1: Provide the entire table in the same page.
Conclusions:
- Line 331: Authors should change haven´t by have not.
Author Response
Thanks so much for the reviewer’s kind suggestion and comments. According to your instructive suggestion, this manuscript has been revised carefully. The followings are our point-by-point responses.
- Line 125: Why authors used the Duncan test instead of Tukey one?
Response: Compared with Tukey test, Duncan test is for comparing one sample (“control”) to each of the others, which proves most efficient for the given data in this article. Duncan test is much more suitable for my experimental data.
- Lines 131-132: please add reference after H-O-H.
Response: As reviewer suggested, the reference has been added after H-O-H at line 133 as listed in references [36].
- Table 1: Provide the entire table in the same page.
Response: The entire table 1 has been provided on the same page.
- Line 331: Authors should change haven´t by have not.
Response: We have changed the "haven't" to "have not" at line 333.